# Men's and women's experiences of violence and traumatic events in rural Côte d'Ivoire before, during and after a period of armed conflict

Mazeda Hossain,[1] Cathy Zimmerman,[1] Ligia Kiss,[1] Drissa Kone,[2] Monika Bakayoko-Topolska,[2] David Manan K A,[2] Heidi Lehmann,[2] Charlotte Watts[1]

[1]London School of Hygiene & Tropical Medicine, London, UK
[2]The International Rescue Committee (IRC), New York, NY, USA

Correspondence to
Mazeda Hossain;
Mazeda.Hossain@lshtm.ac.uk

## ABSTRACT

**Objective:** We assessed men's and women's experiences of gender based violence and other traumatic events in Côte d'Ivoire, a West African conflict-affected setting, before, during and after a period of active armed conflict (2000–2007).

**Design:** Cross-sectional, household survey.

**Setting:** 12 rural communities directly impacted by the Crisis in Côte d'Ivoire, spanning regions controlled by government forces, rebels and UN peacekeepers in 2008.

**Participants:** 2678 men and women aged 15–49 years.

**Primary outcome measures:** Violence exposures measured since age 15. Questions included intimate partner physical and sexual violence; physical and sexual violence by others (including combatants) and exposure to traumatic events before, during and after the Crisis period (2000–2007).

**Results:** Physical and/or sexual violence since age 15 was reported by 57.1% women and 40.2% men (p=0.01); 29.9% women and 12.3% men reported exposure to any violence in the past year. Nearly 1 in 10 women (9.9%) and 5.9% men (p=0.03) were forced to have sex by a non-partner since age 15, and 14.8% women and 3.3% men (p=0.00) reported their first sexual experience was forced. Combatants were rarely reported as sexual violence perpetrators (0.3% women). After the Crisis, intimate partner physical violence was the most frequently reported form of violence and highest among women (20.9% women, 9.9% men, p=0.00). Fearing for their life was reported by men and women before, during and after the Crisis.

**Conclusions:** Sexual violence in conflict remains a critical international policy concern. However, men and women experience different types of violence before, during and after conflict. In many conflict settings, other forms of violence, including intimate partner violence, may be more widespread than conflict-related sexual violence. Alongside service provision for rape survivors, our findings underscore the need for postconflict reconstruction efforts to invest in programmes to prevent and respond to intimate partner violence and trauma.

## Strengths and limitations of this study

- This study presents the first violence and trauma prevalence data over conflict periods from regions spanning rebel, government and UN-controlled forces in Côte d'Ivoire.
- This study addresses the limited population-level data on the various types and severity of gender based violence experienced in a West African setting between women and men.
- Sexual violence figures should be interpreted with caution as we did not explore the broader range of sexual abuse and response bias is possible.

## INTRODUCTION

The past decade has seen unprecedented recognition of sexual violence (SV) in conflict.[1] The UN Security Council alone has issued nine resolutions focused on SV in conflict and fragile state settings since 2000.[2] At the start of its G8 presidency in 2013, the UK announced its firm commitment to address violence against women and girls and in April launched the *G8 Declaration on Preventing Sexual Violence* with a commitment of £23 million by the G8 nations. In the same year, the UK Foreign Office also announced the *Preventing Sexual Violence Initiative*,[3] towards which the UK Government dedicated £10 million to end sexual violence in conflict.[4] However, there is limited evidence on the prevalence and patterns of violence in conflict-affected settings making it difficult for governments, humanitarian and donor agencies to determine how to target their resources most effectively.[5]

Not surprisingly, robust national-level data on the extent of sexual violence are extremely difficult to compile, with current prevalence estimates ranging widely. For example, in the Democratic Republic of Congo (DRC),

reports on the extent of conflict-related sexual violence range from 17.8% to 39.7% among women and 4% to 23.6% among men, due, in part, to methodological differences.[6–8] In the same setting, women also report high levels of violence by an intimate partner (termed intimate partner violence (IPV) or domestic violence), with 35.3% of ever-partnered women reporting sexual partner violence and 56.9% reporting physical partner violence.[9] Recently, data have emerged from Liberia showing high levels of violence and trauma especially among women.[10]

Such data have led to the growing recognition that rape in war is one of numerous forms of violence in conflict-affected settings. Sexual and physical intimate partner violence, child sexual abuse, forced marriage, sexual harassment and rape by non-combatants are also of major concern.[11–14] The most recent Human Security Report (2012) highlighted the discrepancy between the evidence and the international focus on sexual violence in conflict, citing, in particular, evidence that domestic sexual violence may be more prevalent than non-partner rape.[12 15–17]

Côte d'Ivoire is a West African country that has experienced a protracted conflict, known as *the Crisis*, since a coup d'état in 1999.[18–20] In 2002, a UN-French-controlled buffer zone was created, effectively dividing the nation into a rebel-controlled north and a government-controlled south.[21] The first steps of a peace agreement were brokered in 2007, which was followed by a year of limited conflict-related violence between 2007 and 2008. In 2008, however, the country again entered a period of instability before transitioning to an elected President in 2011.[22] Côte d'Ivoire, once considered the 'jewel of West Africa', remains a critical country for regional West African security as it maintains deep ties to neighbouring countries (Mali, Burkina Faso, Ghana, Guinea and Liberia) and other West African nations (Togo, Benin, Sierra Leone and Niger) through migration, trade and remittances. The impact of over a decade of instability and violence is still not known.[23] As the country transitions to the post-conflict period, an understanding of the types of violence and trauma exposures in Côte d'Ivoire may provide insights into programming for health, legal and social sectors within the country and in neighbouring countries such as Mali, which are currently experiencing similar low-level ethnic tensions.[24]

This paper presents the findings from a household survey on men's and women's exposures to interpersonal violence and trauma in 12 rural villages across six administrative districts in Côte d'Ivoire, *prior to* (pre-1999), *during* (2000–2007) and 1 year *after* a period of active conflict (2008).

## METHODS
### Study design and sample
A cross-sectional community survey was conducted in November–December 2008 among 2684 respondents (53% women, 47% men) aged 15–49 years in 12 rural communities across six administrative districts in Côte d'Ivoire. These regions included: Yamoussoukro, Daloa, Bouaflé, Bangolo, Danané and Duekoué. This survey was carried out as a prevalence study prior to the baseline survey of a cluster randomised trial to evaluate the impact of an IPV prevention intervention implemented by a humanitarian organisation in Côte d'Ivoire. The prevalence findings were used to inform the intervention design, which was implemented between 2010 and 2012 in the same communities. (Intervention trial results presented elsewhere).

The study communities were purposively selected based on their accessibility and current relationship with the humanitarian organisation, the International Rescue Committee (IRC). All communities shared similar socioeconomic and population size profiles, with residents relying primarily on agriculture as their main income source. The administrative districts spanned regions controlled by the government, rebels or UN peacekeepers.

Within each community, a representative sample was obtained by first mapping all households to create a sampling list of individuals. All households within each community were eligible to participate. Owing to ethical and safety concerns related to disclosure, we did not aim to interview male and female respondents in the same household. Instead, half of the households in the community were randomly allocated to be 'male' respondent households, and the remainder to be 'female' respondent households. In each household, all eligible household members of the same sex who met the eligibility criteria (15–49 years old and resident in the community for at least 1 year) were invited to participate and be interviewed in private by an interviewer of the same sex. The mapping found that 12 041 individuals lived in the 12 study communities, of which 3471 were eligible to participate and 2869 adults completed an interview (83% response rate). Non-response was generally attributed to being out of town, illness or work reasons.

### Violence and trauma measures
Respondents were asked about their experiences of various acts of violence perpetrated either by an intimate partner (termed 'IPV') or by other perpetrators, such as neighbours, relatives, teachers and combatants (termed 'non-partner violence', NPV).

The IPV module drew on survey instruments used internationally to study IPV.[25–27] All ever-partnered participants were asked: 'Has your partner ever...' perpetrated a specific act of violence, and if so, when (past 12 months, before the past 12 months) and how often (never, sometimes, often) for each time period. Physical violence acts included being: (1) slapped, pushed or (2) hit with something that could hurt you. Severe physical acts were measured by affirmative reports of being (1) kicked, dragged, beaten, (2) choked, burned or (3) threatened with a weapon. Women were also asked about their experiences of partner SV, and specifically

were asked whether they had been (1) physically forced to have sex or (2) forced to have sex due to fear.

An individual was considered to have experienced physical partner violence if she/he reported more than one experience of the following acts: hit with a fist or something else, slapped or had something that could hurt thrown at her/him, pushed or shoved or one act of severe physical violence act. Severe physical violence was defined as reporting at least one experience of the following: kicked or dragged; choked or burned; threatened with a weapon. This categorisation of partner physical violence reflects not only the relative severity of the act (eg, the difference between being slapped vs choked) but also the frequency that the act occurred. It also reflects a more conservative approach that removes individuals (men and women) from the overall prevalence who have experienced a single act of violence, such as a slap. A woman was considered to have experienced sexual partner violence if she reported one or more experiences of forced sex.

Experiences of non-partner violence measured physical violence and sexual violence as an adult (≥15years old). For physical violence, questions included: "Since the age of 15, apart from your partner, has anyone ever physically hurt you?" Sexual violence was measured by: "Since the age of 15, apart from your partner, has anyone ever forced you to have sex against your will?" For both types, follow-up questions were asked about the perpetrator and the timing of the assault in relation to the Crisis (before, during and/or after).

Traumatic events were measured by drawing on the seven domains designated by the Harvard Trauma Questionnaire as common experiences among war-affected populations.[28] Questions were modified to reflect the traumatic experiences potentially associated with conflict-related violence in Côte d'Ivoire. All participants were asked whether they had experienced specific events including: feared for your life, village attacked, witnessed family members seriously hurt/killed, forced to work for someone who attacked your village, forced to have sex with someone who attacked your village, forced to flee your village, family members threatened, seriously hurt by an act of violence, forced to use a weapon against someone, seriously hurt someone and forced to kill someone in defence. A binary variable was created to capture participants who had experienced above median number of events (5 or more).

In each case, respondents who reported affirmatively to having experienced traumatic events were asked about the timing: in the past 12 months (2007–2008; after the Crisis), during the Crisis period (1999–2007; during the Crisis) or before the coup d'état (pre-1999; before the Crisis). To improve recall of event timing, questions were presented with the years and pivotal historical events such as 'before the coup d'état', 'during the time of Gbabgo' or 'during the Crisis' and 'this year', along with the corresponding years.

## Translation, ethics and data collection procedures

The questionnaire was developed in English and French and then translated and back-translated into eight local Ivorian languages. An intensive group translation method was developed by the LSHTM research team where local language speakers translated questions individually and then met as a group (5–10 people) to reach a consensus on the local language interpretation. This interpretation was then checked with the study team and other language groups to ensure that the appropriate and similar meaning was captured across the multiple translations. The final instrument underwent another round of pilot testing and further revision before implementation.

Strict ethical procedures were adopted that recognised possible trauma experienced by the study population and the possibility of renewed violence in the study communities or against field researchers. To ensure the safety of all participants and researchers, all interviewers participated in an intensive 3-week training which included ethical and safety training, and all participants were provided access to psychological and medical support.

Face-to-face interviews were conducted in French or local language in a private setting to reduce levels of bias and improve disclosure. Prior to the start of the interviews, consent for the research project was obtained from village leaders, household heads and individual participants. Quality control measures included the use of multiple checks during the data collection phase and later double-data entry procedures by the data entry team.

## Statistical analysis

The data were double-entered and analysis was completed using Stata V.11.[29] Descriptive data analysis was performed using the Stata survey module. Final analysis was conducted among completed questionnaires. Prevalence data and 95% CIs were calculated using survey commands to account for clustering at the village level. The design effect due to cluster sampling was assessed using Stata V.11 (physical IPV past 12 months, intraclass correlation coefficients (ICC)=0.04 women; ICC=0.03 men). Bivariate and subgroup comparisons between women and men were calculated using the Wald test, where p<0.05 was considered statistically significant. Weighted Demographic Health Survey data[30] from the same study regions were examined to compare the representativeness of the study population against a nationally representative population.

## RESULTS

### Study population

The majority of study participants were under the age of 30, with approximately one-quarter of women (23.7%) and men (22.6%) between 15 and 19 years (table 1). Lower literacy levels were reported by women (31%) than men (59%), and less than half of all participants reported basic literacy levels (44%). Most women (87%)

**Table 1** Weighted study population demographics and comparison with regional figures

| Characteristics | Violence survey data (%) | | Comparative DHS data from same study regions (%) | |
|---|---|---|---|---|
| | Women | Men | Women | Men |
| Age range (years) | | | | |
| 15–19 | 23.7 | 22.6 | 24.0 | 23.3 |
| 20–24 | 20.9 | 16.8 | 19.9 | 20.7 |
| 25–29 | 17.8 | 15.2 | 16.4 | 13.7 |
| 30–34 | 14.0 | 15.7 | 12.6 | 13.0 |
| 35–39 | 10.0 | 13.4 | 10.3 | 11.0 |
| 40–44 | 8.6 | 9.3 | 9.9 | 9.3 |
| 45–49 | 5.0 | 7.0 | 6.9 | 9.1 |
| Total N | 1411 | 1265 | 1407 | 1110 |
| Highest educational attainment | | | | |
| Primary | 19.2 | 24.0 | 30.6 | 31.0 |
| Secondary | 11.1 | 29.9 | 16.1 | 35.5 |
| Higher | 0.1 | 1.6 | 1.0 | 3.7 |
| Not reported/no schooling | 69.6 | 42.9 | 52.3 | 29.8 |
| Total N | 1413 | 1265 | 1407 | 1110 |
| Current living and partnership status | | | | |
| Living with partner (married/boyfriend/girlfriend) | 63.6 | 51.8 | 50.3 | 39.5 |
| Not living with partner (married/boyfriend/girlfriend) | 23.6 | 24.8 | 15.0 | 9.1 |
| No partner reported | 9.5 | 21.8 | 34.7 | 51.3 |
| Not reported | 3.3 | 1.6 | 0.0 | 0.0 |
| Total N | 1413 | 1265 | 1407 | 1110 |
| Religion | | | | |
| Catholic | 13.8 | 11.2 | 17.8 | 18.0 |
| Protestant/evangelical/other Christian | 33.5 | 22.1 | 32.5 | 24.9 |
| Muslim | 18.2 | 18.7 | 22.9 | 21.6 |
| Animist (traditional) | 8.6 | 30.2 | 25.2 | 33.8 |
| Other/none reported | 18.2 | 12.9 | 1.6 | 1.6 |
| Total N | 1412 | 1265 | 1407 | 1110 |
| Number of children | | | | |
| None | 21.5 | 45.4 | 25.0 | 46.4 |
| 1–3 | 45.2 | 33.2 | 39.4 | 27.5 |
| 4–6 | 25.0 | 14.7 | 22.6 | 15.7 |
| 7–9 | 7.2 | 4.8 | 10.1 | 7.2 |
| 10–16 | 1.2 | 2.0 | 2.9 | 3.1 |
| 17+ | 0 | 0 | 0 | 0.2 |
| Total N | 1413 | 1264 | 1407 | 1110 |
| Population in conflict-affected zones | | | | |
| Rebel controlled | 28.5 | 30.2 | 29.1 | 31.3 |
| UN protected | 43.9 | 42.9 | 36.9 | 38.7 |
| National army controlled | 27.7 | 26.9 | 34.0 | 30.0 |
| Total N | 1413 | 1265 | 1407 | 1110 |
| Traumatic conflict-related events | | | | |
| 4 or less experiences | 75.5 | 82.5 | n/d | n/d |
| 5 or more experiences | 21.5 | 17.5 | | |
| Total N | 1412 | 1263 | | |

said they were in a current relationship with a male partner (ie, husband, boyfriend), and over half were cohabiting (63.6%). Most men (77%) also reported having a current relationship with either a wife or girlfriend, with half (51.8%) living with their female partner. Almost one-third (29%) of partnered women and 12% of partnered men reported their relationship was polygamous, in which the male partner had more than one concurrent wife (table 1). Comparisons with the 2005 DHS data suggest that the study population surveyed were representative of the regional population, with similar age breakdowns, religion, number of children and percentage of population living in conflict-affected zones. Differences between partnership status and educational attainment are likely attributable to different definitions between the surveys (table 1).

**Table 2** Prevalence of exposures to physical and sexual interpersonal violence, by sex

| | Prevalence | | | | | | |
|---|---|---|---|---|---|---|---|
| | Women | | | Men | | | |
| Violence type | Per cent | (95% CI) | N | Per cent | (95% CI) | N | p Value* |
| All violence (any perpetrator, all respondents) | | | | | | | |
| Sexual violence | | | | | | | |
| Since age 15 | 32.9 | (28.1 to 38.1) | 1408 | 5.9 | (3.6 to 9.6) | 1256 | 0.00 |
| After the Crisis / Last 12 months | 15.1 | (11.9 to 19.0) | 1408 | 0.1 | (0.0 to 0.8) | 1256 | 0.00 |
| Physical violence | | | | | | | |
| Since age 15 | 47.6 | (41.9 to 53.4) | 1413 | 38.0 | (29.7 to 47.2) | 1265 | 0.05 |
| After the Crisis / Last 12 months | 21.2 | (16.0 to 27.6) | 1413 | 12.2 | (8.9 to 16.4) | 1265 | 0.00 |
| Physical and/or sexual | | | | | | | |
| Since age 15 | 57.1 | (51.7 to 62.3) | 1413 | 40.2† | (31.0 to 50.0) | 1265 | 0.01 |
| After the Crisis / Last 12 months | 29.9 | (25.3 to 34.9) | 1413 | 12.3† | (8.9 to 16.6) | 1265 | 0.00 |
| Intimate partner violence (among ever-partnered) | | | | | | | |
| Sexual violence | | | | | | | |
| Lifetime | 29.1 | (22.3 to 36.9) | 1339 | n/d | n/d | n/d | n/d |
| After the Crisis / Last 12 months | 14.9 | (11.5 to 19.2) | 1332 | n/d | n/d | n/d | n/d |
| Physical violence (any) | | | | | | | |
| Lifetime | 38.4 | (31.7 to 45.5) | 1337 | 19.8 | (12.8 to 29.4) | 1119 | 0.00 |
| After the Crisis / Last 12 months | 20.9 | (15.5 to 27.7) | 1339 | 9.9 | (6.8 to 14.3) | 1120 | 0.00 |
| Physical violence (severe violence) | | | | | | | |
| Lifetime | 23.9 | (18.2 to 30.8) | 1337 | 9.3 | (5.9 to 14.4) | 1119 | 0.00 |
| After the Crisis / Last 12 months | 11.6 | (6.8 to 19.1) | 1339 | 4.2 | (2.7 to 6.5) | 1120 | 0.03 |
| Physical and/or sexual | | | | | | | |
| Lifetime | 49.8 | (42.3 to 57.4) | 1339 | n/d | n/d | n/d | n/d |
| After the Crisis / Last 12 months | 29.7 | (24.9 to 35.0) | 1339 | n/d | n/d | n/d | n/d |
| Non-partner violence (among all respondents) | | | | | | | |
| Sexual violence | | | | | | | |
| Since age 15 | 9.9 | (7.1 to 13.8) | 1408 | 5.9 | (3.6 to 9.6) | 1256 | 0.03 |
| After the Crisis / Last 12 months | 1.1 | (0.6 to 1.8) | 1408 | 0.1 | (0.0 to 0.8) | 1256 | 0.01 |
| Physical violence | | | | | | | |
| Since age 15 | 23.7 | (18.4 to 29.9) | 1412 | 27.1 | (19.9 to 35.5) | 1257 | 0.42 |
| After the Crisis / Last 12 months | 1.9 | (1.3 to 3.2) | 1412 | 3.6 | (2.6 to 4.9) | 1257 | 0.02 |
| Physical and/or sexual | | | | | | | |
| Since age 15 | 27.7 | (22.1 to 34.0) | 1413 | 29.9 | (22.3 to 38.7) | 1265 | 0.60 |
| After the Crisis / Last 12 months | 3.0 | (1.8 to 4.8) | 1413 | 3.6 | (2.6 to 5.0) | 1265 | 0.42 |
| Child Sexual Abuse (among all respondents) | | | | | | | |
| Yes | 7.3 | (4.9 to 10.8) | 1413 | 3.3 | (1.7 to 6.5) | 1265 | 0.04 |
| First sex forced (among all respondents) | | | | | | | |
| Yes | 14.8 | (12.0 to 18.0) | 1333 | 3.3 | (2.4 to 4.5) | 1135 | 0.00 |

All statistics are weighted percentages. Denominators are the sum of the survey weights in the subpopulations of men and women.
*p Value denotes difference between men and women.
†Does not include sexual violence by an intimate partner.
n/d, data not available.

## Prevalence of sexual and physical violence exposures

More than half of all women (57.1%) and over one-third of all men (40.2%) reported an experience of physical and/or SV since age 15 (table 2). Approximately one-third of women (29.9%) and 12.3% of men reported physical and/or SV in the 12-month period following the Crisis. The reported levels of physical and/or SV by non-partners were very similar between men and women, with 27.7% of women and 29.9% of men reporting violence by a non-partner since age 15, and 3% and 3.6% of men and women, respectively, reporting abuse in the past year. This suggests that the difference in the overall levels of violence exposure between the sexes may be attributed to the differing levels of IPV experienced by men and women.

Almost 1 in 10 women (9.9%) reported being forced to have sex by someone other than their intimate partner since age 15, with 1.1% reporting non-partner SV in the past year. For men, the figures were lower but not negligible (5.9% since age 15, 0.1% in the past year). Many women also reported forced sex by a partner, with 29.1% and 14.9% of ever-partnered women reporting forced sex ever, and in the past year (table 2). In combination, these figures suggest that 32.9% of women have experienced SV since age 15, with most of this SV (24% overall) being perpetrated by their intimate partners, and with 5.9% of

women reporting sexual assault by both an intimate partner and other men. Importantly, 14.8% of women and 3.3% of men reported that their first sexual experience was forced (table 2).

Nearly half of women and over one-third of men reported having one or more experiences of physical violence since the age of 15 years old (47.6% women, 38% for men). The levels of physical violence by non-partners were very similar between the sexes over their lifetime (27.7% women, 29.9% men), and in the 12 months following the peace agreement (3% women, 3.6% men). In contrast, the reported levels of physical violence by a partner after the Crisis were more than twice as high for women compared with men (20.9% vs 9.9%), with women also being more likely to report experiencing severe acts of physical violence by a partner compared with men (23.9% women, 9.3% men, p=0.00) in their lifetime (table 2).

### Perpetrators

Respondents reported a broad range of physical and SV perpetrators. Table 3 presents the prevalence of non-partner sexual and physical violence perpetrators overall, and broken down according to whether the violence occurred before, during or after the Crisis periods.

Nearly 1 in 10 women (9.9%) reported SV perpetrated by someone other than their partner, with SV most often perpetrated by male strangers or acquaintances. Only a small percentage of women reported SV perpetrated by a combatant (0.3%). Among men reporting SV from someone other than an intimate partner (5.9%), the most common perpetrators were female acquaintances (3.4% overall) and female strangers (1.8%; table 3).

The reported prevalence of non-partner SV was lower after the Crisis period than during or before the Crisis period, potentially as a result of the difference in length of time reflected in each measure. In contrast, the prevalence of SV by an intimate partner remained high (14.9% among women after the Crisis; table 2).

Men reported experiencing higher levels of physical violence during the Crisis than women (8.9% women, 12.6% men, p=0.02). The perpetrators who were typically named were family members for men and women, except during the Crisis period, when men were more likely than women to report physical assault from combatants (0.9% women, 4.7% men, p=0.00).

### Exposure to traumatic conflict-related events

'Feared for your life' was the most common traumatic event reported, with nearly all participants acknowledging having had at least one experience when they feared for their life since age 15 (90% women, 83% men). As expected, levels of all trauma exposures were higher during the active conflict period. Little difference was noted between men and women, except for fearing for one's life, which was higher among women at all time periods. Among all participants, 19.6% reported experiencing five or more traumatic events in their lifetime (figure 1).

## DISCUSSION

As the international community has intensified its focus on sexual violence against women in conflict settings, our findings from Côte d'Ivoire support increased policy and programming attention to all forms of violence in conflict-affected countries, particularly domestic violence against women. Findings from war-affected contexts show that both sexes are subjected to various forms of abuse. However, our data indicate that when resources are limited, a focus on preventing violence against women and girls (whether through direct services for survivors or primary prevention efforts) is important, as women experience violence in significantly greater proportions and are often exposed to more severe abuses, especially in the post-conflict period. In our study, women experienced the highest levels of violence within and outside of their homes, and were most likely to report the most severe forms of physical violence by a partner (ie, dragged, kicked, choked), in addition to experiencing SV by intimate partners and non-partners.

Our findings also confirm that attention to conflict-related SV is warranted. Yet, at the same time, these results emphasise that focusing narrowly on rape in war in all conflict-affected settings is short-sighted. Our data strongly indicate that violence occurs in many forms and is perpetrated by different individuals, in addition to combatants. The most common perpetrators of violence against women in our study appear to be intimate partners, family members and acquaintances, while men report violence from family members, acquaintances, and, during the conflict period, from combatants.

Strategies should not, however, exclude programming which responds to violence against men. Importantly, these findings highlight that men are also victims of multiple forms of abuse, including SV. Furthermore, there is reason to believe that men in conflict settings who have experienced violence, especially SV, are likely to have little support and may be less likely to disclose. Men reported higher levels of non-partner physical violence experiences during and after the Crisis. Non-partner SV was higher among women.

This study is not nationally representative of Côte d'Ivoire and covers a subsection of the country. In addition, we only measured forced sex and did not explore the broader range of forms of sexual abuse that may occur. We prioritised comparability between men and women and used survey questions that are more widely used among women.[31] However, we did not pose questions on sexual IPV to men, and at the time of implementation, there was little research on female-to-male sexual abuse, making data interpretation difficult without a more in-depth understanding of the phenomena. Therefore, the SV prevalence figures should be interpreted with caution, as it is unclear how comparable the data are for men and women. In our study, men identified females who were friends and strangers as perpetrators; however, data was not collected on the nature of these relationships, nor the trauma that may

**Table 3** Prevalence of non-partner sexual and physical violence since age 15, by perpetrator types, conflict time period and sex

| Non-partner violence perpetrators | Before Crisis (pre-1999) | | | During Crisis (2000–2007) | | | After Crisis (2007–2008) | | | Lifetime | | |
|---|---|---|---|---|---|---|---|---|---|---|---|---|
| **SEXUAL VIOLENCE** (>15 years old) *Forced or coerced sex* | **Women (%)** (n=1408) | **Men (%)** (n=1256) | **p Value*** | **Women (%)** (n=1408) | **Men (%)** (n=1256) | **p Value*** | **Women (%)** (n=1408) | **Men (%)** (n=1256) | **p Value*** | **Women (%)** (n=1408) | **Men (%)** (n=1256) | **p Value*** |
| Any perpetrator (male/female) | 5.4 | 3.6 | 0.36 | 4.0 | 2.2 | 0.20 | 1.1 | 0.1 | 0.11 | 9.9 | 5.9 | 0.09 |
| Family member (male)* | 0.6 | 0.0 | 0.25 | 0.6 | 0.1 | 0.26 | 0.0 | 0.0 | 0.52 | 1.1 | 0.1 | 0.11 |
| Family member (female)* | 0.1 | 0.2 | 0.48 | 0.1 | 0.2 | 0.46 | 0.0 | 0.0 | 0.52 | 0.1 | 0.5 | 0.35 |
| Acquaintance (male) | 1.2 | 0.2 | 0.19 | 0.8 | 0.2 | 0.24 | 0.1 | 0.0 | 0.43 | 2.0 | 0.4 | 0.08 |
| Acquaintance (female) | 0.2 | 2.2 | 0.02 | 0.1 | 1.3 | 0.06 | 0.0 | 0.3 | 0.08 | 0.3 | 3.4 | 0.00 |
| Acquaintance (sex unknown) | 0.1 | 0.0 | 0.44 | 0.1 | 0.0 | 0.54 | 0.0 | 0.0 | 0.52 | 0.2 | 0.0 | 0.37 |
| Stranger/other not identified (male) | 2.6 | 0.0 | 0.02 | 1.9 | 0.1 | 0.03 | 0.2 | 0.0 | 0.38 | 4.5 | 0.1 | 0.00 |
| Stranger/other not identified (female) | 0.8 | 1.1 | 0.58 | 0.4 | 0.6 | 0.58 | 0.1 | 0.2 | 0.57 | 1.1 | 1.8 | 0.41 |
| Stranger/other not identified (sex unknown) | 0.0 | 0.0 | 0.52 | 0.1 | 0.0 | 0.45 | 0.0 | 0.0 | 0.52 | 0.1 | 0.0 | 0.45 |
| Combatant/uniformed official | 0.4 | 0.7 | 0.52 | 0.3 | 0.2 | 0.68 | 0.1 | 0.0 | 0.53 | 0.3 | 0.2 | 0.68 |
| **PHYSICAL VIOLENCE** (>15 years old) *Physically mistreated or hit* | **Women (%)** (n=1412) | **Men (%)** (n=1265) | **p Value*** | **Women (%)** (n=1412) | **Men (%)** (n=1265) | **p Value*** | **Women (%)** (n=1412) | **Men (%)** (n=1265) | **p Value*** | **Women (%)** (n=1412) | **Men (%)** (n=1265) | **p Value*** |
| Any perpetrator (male/female) | 15.4 | 15.2 | 0.36 | 8.9 | 12.6 | 0.13 | 1.9 | 3.6 | 0.02 | 23.6 | 26.9 | 0.27 |
| Family member (male) | 6.4 | 8.7 | 0.09 | 3.2 | 3.7 | 0.18 | 0.9 | 1.1 | 0.10 | 8.9 | 12.9 | 0.03 |
| Family member (female) | 8.2 | 2.5 | 0.00 | 4.5 | 1.5 | 0.00 | 0.9 | 1.1 | 0.16 | 8.9 | 3.7 | 0.01 |
| Acquaintance (male) | 1.6 | 2.4 | 0.23 | 0.7 | 1.4 | 0.15 | 0.2 | 0.8 | 0.04 | 2.1 | 4.4 | 0.09 |
| Acquaintance (female) | 0.9 | 0.0 | 0.03 | 0.4 | 0.2 | 0.09 | 0.1 | 0.0 | 0.08 | 1.1 | 0.4 | 0.04 |
| Acquaintance (sex unknown) | 0.3 | 1.7 | 0.01 | 0.0 | 1.2 | 0.01 | 0.0 | 0.6 | 0.01 | 0.3 | 2.9 | 0.00 |
| Stranger/other not identified (male) | 1.3 | 2.6 | 0.01 | 0.9 | 1.8 | 0.04 | 0.2 | 0.6 | 0.18 | 2.3 | 4.6 | 0.01 |
| Stranger/other not identified (female) | 0.2 | 0.2 | 0.10 | 0.0 | 0.2 | 0.05 | 0.1 | 0.6 | 0.07 | 0.2 | 0.3 | 0.08 |
| Stranger/other not identified (sex unknown) | 0.1 | 0.0 | 0.10 | 0.0 | 0.2 | 0.11 | 0.1 | 0.1 | 0.15 | 0.1 | 0.2 | 0.20 |
| Combatant/uniformed official | 0.0 | 0.6 | 0.04 | 0.9 | 4.7 | 0.00 | 0.0 | 0.2 | 0.61 | 0.9 | 5.3 | 0.00 |

'All statistics are weighted percentages. Denominators are the sum of the survey weights in the subpopulations of men and women.
Family includes: father/mother, father/mother-in-law and other family members; acquaintances include: friends, family friends, neighbours, teachers, religious leaders; strangers/others include: strangers, individuals not identified; combatant/uniformed official includes: someone who attacked your village, uniformed official (ie, police, gendarme, military), sex unspecified.
*p Value denotes difference between men and women.

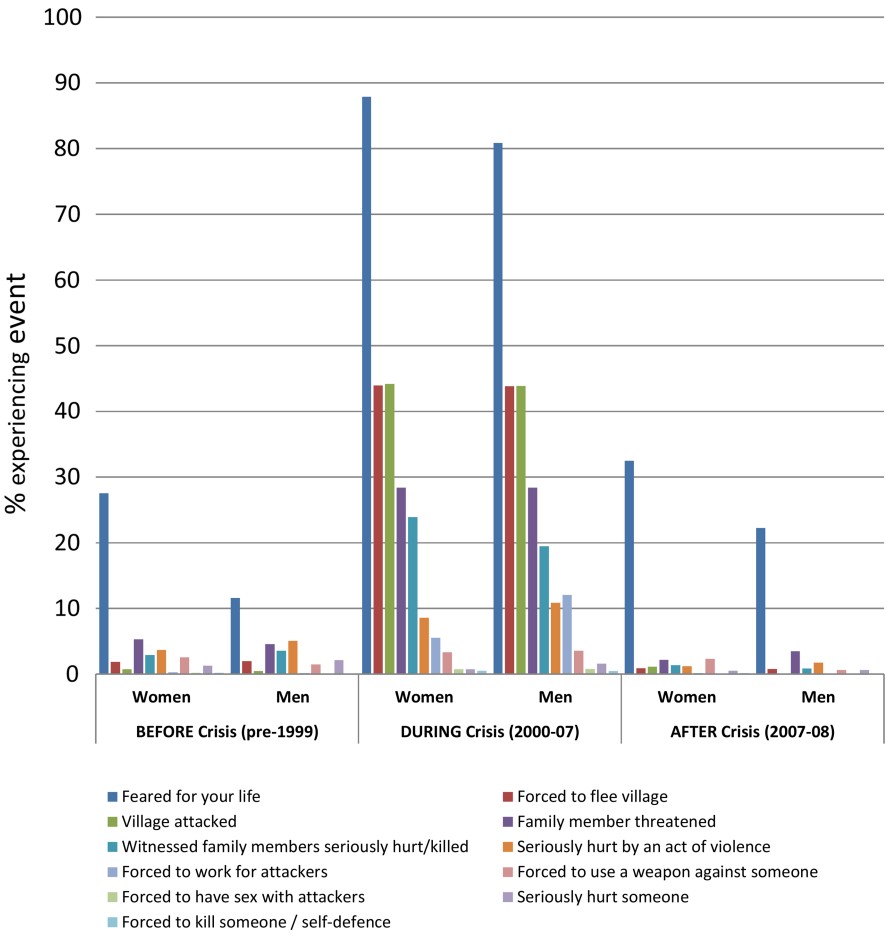

**Figure 1** Prevalence of traumatic experiences by type of event, timing in relation to the Crisis and by sex (women: n=1412; men: n=1263).

have resulted from being forced to have sex. Other research suggests that for men, being forced to have sex by a woman may have different implications than for women who are victims of forced sex and is an area that requires further research.[32] There is also the potential for response bias as given the sensitive nature of the questions participants may have been reluctant to report forced sex. Furthermore, although no remuneration was given for participation in the study, there remains the possibility those respondents over-reported, or under-reported, in hopes of receiving services.

Given the range of violence detected in conflict-affected communities, our findings pose a significant challenge to national governments and the international community. To truly make inroads in reducing violence against women, programming must address SV against women *in conjunction with* the many other types of violence that occur to men and women. SV, as our data demonstrate, does not occur in isolation. Indeed, especially in contexts where violence is widespread, such as war-torn areas, forms of violence are likely to be interrelated, potentially exacerbating one another.

Moreover, strategies to address violence occurring in a conflict-affected setting, where so many individuals have been exposed to additional traumatic events over the course of their lives, will also need to consider how this range of psychologically damaging circumstances might

influence intervention efforts. For example, a majority of study participants reported that they had 'feared for their life', and over 40% of respondents reported being forced to flee their villages due to a violent attack. Promoting recovery and behaviour change (reducing levels of IPV) in a context of fear will undoubtedly be challenging.

Post-conflict reconstruction efforts within West Africa have traditionally focused on security, physical infrastructure and economic development rather than gendered human security issues.[33] However, as Côte d'Ivoire and its neighbour's transition towards peace, the issue of violence against women cannot be ignored. For decision-makers and programmes that have the explicit aim of addressing violence against women in the longer term, this study provides, what we hope will be, the beginning of a growing evidence base to foster comprehensive, gender-informed strategies to improve the safety, health and well-being of men, women and children in conflict-affected settings.

**Acknowledgement** The authors are grateful to all the study respondents who participated in the study. They also gratefully acknowledge the dedicated commitment of International Rescue Committee (IRC) country staff, the research field staff and C Burkholder.

**Contributors** MH, CZ, CW and LK were responsible for the design and conduct of the study. MH and DK were responsible for field management and study instrument development. MT and HL contributed to national-level data collection procedures and oversight. All authors contributed to the development and adaptation of the questionnaires and interpretation of the

data. MH was responsible for statistical data analysis. MH, CZ and CW drafted the manuscript which was reviewed and approved by all authors.

**Funding** This work was supported by funding from the NoVo Foundation, the Sigrid Rausing Trust and the Economic and Social Research Council (ESRC) (ES/J021032/1).

**Competing interests** None.

**Patient consent** Obtained.

**Ethics approval** Ethical approval for this study was received in 2008 from the London School of Hygiene & Tropical Medicine Ethics Committee. Local ethics approval was received from the Ministry of Family, Women and Social Affairs in Côte d'Ivoire.

**Provenance and peer review** Not commissioned; externally peer reviewed.

**Data sharing statement** Preliminary study findings have been presented to the donors, at specialised conferences and several expert group meetings on sexual violence in conflict settings. The data contained in this report are not publically available in a peer-review format.

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
