## [Reviewer comments · BMJ Open]

Some articles will have been accepted based in part or entirely on reviews undertaken for other BMJ Group journals. These will be reproduced where possible.

ARTICLE DETAILS

TITLE (PROVISIONAL)	Sexual and physical violence in a West-African armed conflict-affected setting: Men's and women's experiences of violence and traumatic events in rural Côte d'Ivoire before, during and after a period of armed conflict
AUTHORS	Hossain, Mazedra; Zimmerman, Cathy; Kiss, Ligia; Kone, Drissa; Topolska, Monika; Manan, David; Lehman, Heidi; Watts, Charlotte

VERSION 1 - REVIEW

REVIEWER	Lynn Lawry MD, MSPH, MSc Director, Initiative in Global Women's Health Brigham and Women's Hospital/Harvard Medical School Boston, MA USA Lawry Research Associates International Maryland, USA
REVIEW RETURNED	28-Aug-2013

GENERAL COMMENTS	The authors did not do an extensive search on conflict/nonconflict related SGBV research. They mention a few but miss many seminal publications in major medical journals and social science literature. The statistics are not appropriate for looking at data across varying time periods. This study answers an advocacy question but not a scientific hypothesis. The lack of adequate statistics does not make this credible. Data need weighting and analysis for differing time periods. Conclusions are based on descriptive statistics that may over-represent the real data. Although they discuss previous evidence, they leave out many well known and important work by others (several research groups) in the region and in many other conflicts. Local IRB was not discussed. In all of the studies I have done in the past, local IRB was/is required of all studies. If omitted, this needs to be stated. I applaud IMCs change to recognize male sexual violence however, this is a few steps forward as there was still a bias that men cannot suffer sexual IPV; instruments should have been adapted to assess men as well.
---

Title: Needs to reflect the representativeness of this study. This study is not generalizable to men and women's experience in Cote D'Ivoire but to a limited area. As written and to a policymaker or journalist, this can be taken out of context.

Abstract

Line 6-8 ...limited population data on its magnitude in different settings and little on the prevalence of other serious forms of violence.

This is not a true statement. Studies like this and many national population based studies have been done in many different conflict settings as well as natural disaster settings and report other "serious forms of violence".

Introduction

Lines 13-16 Yet, the limited evidence on the prevalence and patterns of violence in conflict-affected settings makes it difficult for governments, humanitarian and donor agencies to determine how to target their resources most effectively.

Again, this is not a true statement, the research is there but governments and policymakers do not access/use/want to use data that has been collect over decades.

The authors need to do a much better background search and citation on this type of work.

Violence and Trauma

Lines 21-25 Same as above. Agree with part of the citations, however, there are other estimates of IPV in DRC and there are many other studies in the region and continent that assess IPV in Liberia (even before the Liberia study referenced), and other post conflict settings.

Line 45-48, by only asking women these questions, you have biased the study. All questions should be gender neutral. As this appears to be a program evaluation for IRC, I understand the need to have targeted questions to meet your M&E needs, however, surveys of SGBV should not gender specific as it highly biases responses and data. Address in limitations please. Particularly when you go on in results with the following...clearly men had female perpetrators.

"Fewer men (5•9%) reported sexual violence from someone other than their partner, with the most common perpetrators being female acquaintances (3•4% overall), and female strangers (1•8%). (Table 3)"

Page 6 Lines 25-27 Timing and memory. Needs to be discussed as a limitation or at least defend why you think people can remember these timings and violence accurately.

Ethics Review: Most ethics committees require local review. Cote D'Ivoire has several IRBs. Why wasn't the local IRB used as well. The ethics committee needs to be spelled out.

	Statistical Analysis Was the data weighted? By what variables....Why or why not? The time before, during and after are different time periods (years). You have to use a factor to account for the differing numbers of years to control for over or under reporting. Page 10: lines 13-15 ....as women experience violence in significantly greater proportions and are often exposed to more severe abuses. This is an untrue, undocumented and an advocacy statement. Recent work and conferences (see USIP) have shown consistently that men have not been researched and the true magnitude and prevalence of male sexual violence is only beginning to be understood because aid agencies and advocacy groups have used bias research to document SGBV. Furthermore, this is not an ethical justification to focus on women. Using this logic we can say that since only 4% of men suffer from breast cancer, we can ignore them and let them die because “women have breast cancer in great proportions” Again, statement like this point to a limited subject review of background work and prior research.
--	--

REVIEWER	Nancy Glass Professor Johns Hopkins University USA I have no conflict of interest
REVIEW RETURNED	21-Sep-2013

- The reviewer completed the checklist but made no further comments.

VERSION 1 – AUTHOR RESPONSE

Reviewer: Lynn Lawry MD, MSPH, MSc
 Director, Initiative in Global Women's Health
 Brigham and Women's Hospital/Harvard Medical School
 Boston, MA USA

Lawry Research Associates International
 Maryland, USA

1. The authors did not do an extensive search on conflict/nonconflict related SGBV research. They mention a few but miss many sentinel publications in major medical journals and social science literature.

Response 1:

Thank you for your comments. We agree with Dr. Lawry that many other prevalence studies exist

however, our study represents the only research we know of from Côte d'Ivoire which measures interpersonal violence among both women and men in the peer-review literature using comparable measures.

The research cited in the introduction is intended to set the context of existing research on GBV in conflict settings. Studies cited were chosen if they collected population- or household-level data among women and men on violence. We did not include grey literature publications in the citations. We greatly appreciate Dr. Lawry's feedback and would welcome suggestions on additional references to be included, especially those not accessible in the peer-reviewed evidence base.

2. The statistics are not appropriate for looking at data across varying time periods.

Response 2:

Although three time periods are presented, the data was collected in a cross-sectional study. We did not use a longitudinal design for the community survey with measurements taken at three time points. The observations were recorded from the same individuals reporting their experience during three different periods of Ivorian history.

3. This study answers an advocacy question but not a scientific hypothesis.

Response 3:

Thank you for the comment. We hope that the following helps to clarify that the paper presents the results of a cross-sectional prevalence study. At the time of data collection (and to date), there was very little data available on levels of gender based violence (intimate partner and non-partner violence) in Cote d'Ivoire. The research aim was to address this gap by describing the prevalence and severity of interpersonal violence among a rural population in Cote d'Ivoire. The aim of the analysis was descriptive and did not test a hypothesis. We have adhered to BMJ guidelines that support our decision on how to frame our aim and choice of analytical methods.

"In terms of selecting a statistical test, the most important question is "what is the main study hypothesis?" In some cases there is no hypothesis; the investigator just wants to "see what is there". For example, in a prevalence study there is no hypothesis to test, and the size of the study is determined by how accurately the investigator wants to determine the prevalence."

Source:

<http://www.bmj.com/about-bmj/resources-readers/publications/statistics-square-one/13-study-design-and-choosing-statisti>

4. The lack of adequate statistics does not make this credible. Data need weighting and analysis for differing time periods.

Response 4:

Please see our response 3 above.

This was not a longitudinal study with measurements taken at three times points. Rather, observations were recorded from the same individuals reporting their experience during three different periods of Ivorian history (for traumatic events) and two points for IPV and NPV (last 12 months which corresponds to 'after the Crisis' period and before).

Our analysis included checking the design effect of the cluster sampling. We have added the following to the methods section:

"Descriptive data analysis was performed using the Stata survey commands. Final analysis was conducted among completed questionnaires. Prevalence data and 95% confidence intervals were calculated using survey commands to account for clustering at the village level. The design effect due to cluster sampling was assessed using Stata (physical IPV last 12 months ICC=0.04 women; ICC= 0.03 men). Bivariate and sub-group comparisons were calculated using the Wald test where $p < 0.05$

was considered statistically significant.”

5. Conclusions are based on descriptive statistics that may over-represent the real data.

Response 5:

Our study was designed to estimate the prevalence of IPV and other forms of violence among our population of interest. We approached all households within the 12 selected communities to identify respondents and had a high response rate (83%). Given this and the comparison with the DHS population data, we do not believe that our sample is not representative of (or over-represents) our population of interest. We have also presented 95% CIs for our estimated IPV and NPV outcomes.

Our aim was to present descriptive data on the prevalence of violence in the settings described in Côte d'Ivoire. We have also presented DHS data from the same regions as the study sites to check for comparability of the study population.

We feel that one of the strengths of the study is that the IPV questions not only ask women and men about violence from the current or last partner, but also collect data on lifetime experiences of IPV, thereby reducing the under-reporting of estimates as violence from previous partners is captured.

In addition, we have also captured IPV data that reflects the severity and frequency of physical IPV through the creation of a physical IPV variable that does not count one-time events of 'less severe' acts such as slapping or pushing. In other studies measuring IPV, physical IPV acts are typically aggregated together without consideration for the severity of the act (i.e., a one-time push should not necessarily be given the same weight as the act of choking.)

We have modified the methods section with the following:

“This survey was carried out as a prevalence study prior to the baseline survey of a cluster randomised trial to evaluate the impact of an IPV prevention intervention implemented by a humanitarian organisation in Côte d'Ivoire. The prevalence findings were used to inform the intervention design, which was implemented between 2010-2012 in the same communities.”

6. Although they discuss previous evidence, they leave out many well known and important work by others (several research groups) in the region and in many other conflicts.

Response 6:

Please see Response 1. We welcome any feedback the reviewer may have to strengthen the paper.

7. I applaud IMCs change to recognize male sexual violence however, this is a few steps forward as there was still a bias that men cannot suffer sexual IPV; instruments should have been adapted to assess men as well.

Response 7:

We agree with Dr Lawry and LSHTM's subsequent data collection rounds in Côte d'Ivoire included measurements of sexual IPV against men following the collection of qualitative data to further understand the phenomena. (This data will be presented in future publications.) Data on sexual IPV against men was not collected in this 2008 prevalence study. The LSHTM study instrument utilised an adapted version of the instrument used in the WHO Multi-Country Study on Domestic Violence against Women. To ensure comparability, men and women were asked the same questions (except for sexual IPV). All IPV related questions referred to their latest intimate partner, who was identified in the earlier part of the questionnaire.

The discussion section includes the following:

“Other research suggests that for men, being forced to have sex by a woman may have different implications than for women and is an area that requires further research.³² There is also the

potential for response bias as given the sensitive nature of the questions, participants may be reluctant to report forced sex. Furthermore, although no remuneration was given, there remains the possibility those respondents over-reported, or under-reported, in hopes of receiving services.”

8. Local IRB was not discussed. In all of the studies I have done in the past, local IRB was/is required of all studies. If omitted, this needs to be stated.

Response 8:

Thank you. We added the following to the methods section:

“Ethical clearance for the project was obtained from the LSHTM (London) and the Ministry of Family, Women and Social Affairs (Abidjan) in 2008.”

9. Title: Needs to reflect the representativeness of this study. This study is not generalizable to men and women’s experience in Cote D’Ivoire but to a limited area. As written and to a policymaker or journalist, this can be taken out of context.

Response 9:

We appreciate the feedback and have changed the title to include ‘rural’ and ‘armed’ to reflect that the data is drawn from rural sites in Côte d’Ivoire, a nation that has experienced an armed conflict. Included in the paper is a comparative analysis with populations within the same region of Côte d’Ivoire using Demographic Health Survey (DHS) data to check the comparability of the study population. This comparative data is included in Table 1 and explained in the ‘study population’ section description within the results section and further within the limitations section of the discussion.

We have included the following within the results and discussion sections:

“Comparisons with the 2005 DHS data suggests that the study population surveyed are representative of the regional population, with similar age breakdowns, levels of educational attainment, number of children and percentage of population living in conflict-affected zones. Differences between partnership status and religion are likely attributable to different definitions between the surveys (Table 1).”

“This study was limited as it is not nationally representative of Côte d’Ivoire and covers a sub-section of the country.”

10. Abstract: Line 6-8 ...limited population data on its magnitude in different settings and little on the prevalence of other serious forms of violence.

This is not a true statement. Studies like this and many national population based studies have been done in many different conflict settings as well as natural disaster settings and report other “serious forms of violence”.

Response 10:

Compared to thousands of studies collecting research on intimate partner violence against women, we were unable to locate many peer-review studies which collected data not only multiple forms of violence (IPV, NPV and trauma) but also collected it among women and men. We agree with Dr Lawry that similar studies exist but within the IPV evidence base, there are still relatively few population level studies from conflict-affected settings which have published this data. Among existing research, data is often collected from women only or men only or specific sub-populations such as ex-combatants or children. For example, within the DHS surveys, we found only three surveys from conflict-affected settings that collected data on sexual violence (DRC, Liberia, Rwanda), and this was only collected among women. In addition, the last published DHS from Côte d’Ivoire did not collect this range of data (IPV, NPV and traumatic events) from both women and men. However, we acknowledge that much data emerging from conflict settings that is not published within the peer-review literature; therefore we have modified the abstract to focus on the study objective only.

“Objective:

We assessed women’s and men’s experiences of gender based violence and other traumatic events in Cote d’Ivoire, a West African conflict-affected setting before, during, and after a period of active armed conflict (2000-2007).”

11. Introduction: Lines 13-16 Yet, the limited evidence on the prevalence and patterns of violence in conflict-affected settings makes it difficult for governments, humanitarian and donor agencies to determine how to target their resources most effectively.

Again, this is not a true statement, the research is there but governments and policymakers do not access/use/want to use data that has been collect over decades. The authors need to do a much better background search and citation on this type of work.

Response 11:

Please see Response 10.

12. Violence and Trauma: Lines 21-25 Same as above. Agree with part of the citations, however, there are other estimates of IPV in DRC and there are many other studies in the region and continent that assess IPV in Liberia (even before the Liberia study referenced), and other post conflict settings.

Response 12:

Thank you. The paragraphs referred to in Comment 11 and 12 were presented as background to the range of violence in a conflict-affected setting. The IPV examples were taken from a few studies in the DRC to illustrate how methodological differences can affect the prevalence range. The Liberia citation refers only to trauma measured, not IPV, therefore only one citation was included.

“Not surprisingly, robust national level data on the extent of SV are extremely difficult to compile, with current prevalence estimates ranging widely. For example, in the Democratic Republic of Congo (DRC), reports on the extent of conflict-related sexual violence range from 17•8%-39•7% among women and 4%-23.6% among men, due, in part, to methodological differences.6-8 In the same setting, women also report high levels of violence by an intimate partner (termed intimate partner violence (IPV) or domestic violence), with 35•3% of ever-partnered women reporting sexual partner violence and 56•9% reporting physical partner violence.9 More recently, data has emerged from Liberia showing high levels of violence and trauma especially among women.10”

13. Line 45-48, by only asking women these questions, you have biased the study. All questions should be gender neutral. As this appears to be a program evaluation for IRC, I understand the need to have targeted questions to meet your M&E needs, however, surveys of SGBV should not gender specific as it highly biases responses and data. Address in limitations please. Particularly when you go on in results with the following...clearly men had female perpetrators.

Response 13:

Please see Response 7.

All men and women were asked the same questions (except for sexual IPV) with the same wording. IPV related questions referred to their latest intimate partner, who was identified in the earlier part of the questionnaire.

We have modified the limitations section as follows:

“This study was limited as it is not nationally representative of Côte d’Ivoire and covers a sub-section of the country. In addition, we only measured forced sex and did not explore the broader range of forms of sexual abuse that may occur. We prioritised comparability between women and men and used survey questions that are more widely used among women.31 However, we did not pose questions on sexual IPV to men and at the time of implementation there was little research on female-to-male sexual abuse, making data interpretation difficult without a more in-depth understanding of the phenomena. Therefore, the sexual violence prevalence figures should be interpreted with caution, as it is unclear how comparable the data is for women and men. In our study, men identified females

that were friends and strangers as perpetrators, however, data was not collected on the relationship or trauma that may have resulted from being forced to have sex. Other research suggests that for men, being forced to have sex by a woman may have different implications than for women and is an area that requires further research.³² There is also the potential for response bias as given the sensitive nature of the questions, participants may be reluctant to report forced sex. Furthermore, although no remuneration was given, there remains the possibility those respondents over-reported, or under-reported, in hopes of receiving services.”

14. Page 6 Lines 25-27 Timing and memory. Needs to be discussed as a limitation or at least defend why you think people can remember these timings and violence accurately.

Response 14:

Thank you for the comment. We pilot tested several methods and found that the inclusion of key political events worked best to improve timing recall. We have added the following to the methodology:

“To improve recall of event timing, questions were presented with both the years and pivotal historical events such as ‘before the coup d’état’, ‘during the time of Gbabgo’ or ‘during the Crisis’ and ‘this year’, along with the corresponding years.”

15. Ethics Review: Most ethics committees require local review. Cote D’Ivoire has several IRBs. Why wasn’t the local IRB used as well. The ethics committee needs to be spelled out.

Response 15:

Thank you.

We added the following to the methods section:

“Ethical clearance for the project was obtained from the LSHTM (London) and the Ministry of Family, Women and Social Affairs (Abidjan) in 2008.”

16. Statistical Analysis: Was the data weighted? By what variables...Why or why not?

The time before, during and after are different time periods (years). You have to use a factor to account for the differing numbers of years to control for over or under reporting.

Response 16:

Please see Response 4.

17. Page 10: lines 13-15:as women experience violence in significantly greater proportions and are often exposed to more severe abuses.

This is an untrue, undocumented and an advocacy statement. Recent work and conferences (see USIP) have shown consistently that men have not been researched and the true magnitude and prevalence of male sexual violence is only beginning to be understood because aid agencies and advocacy groups have used bias research to document SGBV. Furthermore, this is not an ethical justification to focus on women. Using this logic we can say that since only 4% of men suffer from breast cancer, we can ignore them and let them die because “women have breast cancer in great proportions” Again, statement like this point to a limited subject review of background work and prior research.

Response 17:

We greatly appreciate Dr. Lawry’s viewpoint and feedback. In our study population, women, however, reported higher levels of physical violence from male intimate partners, family members and acquaintances. Severe physical IPV was also reported by a greater proportion of women compared to men. Men reported higher levels of physical violence from family members and acquaintances and combatants.

We do not attempt to posit that violence against men should be ignored however and have been careful to address this need through the following paragraph on addressing men's experiences of violence:

"While it is clear that focussing limited resources specifically on violence against women is justified, strategies should not, however, exclude violence experienced by men. Importantly, these findings highlight that men are also victims of multiple forms of abuse, including sexual violence. Furthermore, there is reason to believe that men in conflict settings who have experienced violence, especially sexual violence, are likely to have little support and may be less likely to disclose. Men reported higher levels of non-partner physical violence experiences during and after the Crisis. Non-partner sexual violence was higher among women."

Reviewer: Nancy Glass
Professor
Johns Hopkins University
USA

I have no conflict of interest

(There are no comments. Publication recommended.)